# Piezoelectric Ring Bender for Characterization of Shear Waves in Compacted Sandy Soils

**DOI:** 10.3390/s21041226

**Published:** 2021-02-09

**Authors:** Dong-Ju Kim, Jung-Doung Yu, Yong-Hoon Byun

**Affiliations:** 1School of Agricultural Civil & Bio-Industrial Engineering, Kyungpook National University, Daegu 41566, Korea; kyrix1028@knu.ac.kr; 2School of Civil, Environmental and Architectural Engineering, Korea University, Seoul 02841, Korea; noorung2@korea.ac.kr

**Keywords:** compacted soil, piezoelectric transducers, shear modulus, shear wave velocity

## Abstract

Shear wave velocity and small-strain shear modulus are widely used as the mechanical properties of soil. The objective of this study is to develop a new shear wave monitoring system using a pair of piezoelectric ring benders (RBs) and to evaluate the suitability of RB in compacted soils compared with the bender element and ultrasonic transducer. The RB is a multilayered piezoelectric actuator, which can generate shear waves without disturbing soils. For five compacted soil specimens, the shear waves are monitored by using three different piezoelectric transducers. Results of time-domain response show that the output signals measured from the RB vary according to the water content of the specimen and the frequency of the input signal. Except at the water content of 9.3%, the difference in the resonant frequencies between the three transducers is not significant. The shear wave velocities for the RB are slightly greater than those for the other transducers. For the RB, the exponential relationship between the shear wave velocity and dry unit weight is better established compared with that of the other transducers. The newly proposed piezoelectric transducer RB may be useful for the evaluation of the shear wave velocity and small-strain shear modulus of compacted soils.

## 1. Introduction

Soils behave in a nonlinear manner and deform in a plastic manner. However, at strain levels below the linear cyclic threshold shear strain of 0.001%, soil behaves essentially as a linear elastic material [1]; the corresponding shear modulus of soil is considered a constant maximum value and is referred to as small-strain shear modulus. The small-strain shear modulus is a key parameter in soil dynamics and earthquake engineering related to the void ratio, effective stress, stress history, degree of saturation, grain characteristics, consolidation, compaction, density, and plastic index of soil [2,3,4,5]. According to elastic theory, the small-strain shear modulus is directly related to shear wave velocity through soil bulk density. Shear waves can only propagate through soil skeleton and induce shear deformation of soil. Therefore, shear wave velocity is regarded as an effective stress parameter [6,7]. Thus, the shear wave velocity and the small-strain shear modulus are widely used as the mechanical properties of soils in geotechnical engineering for the design of foundations under dynamic loading, earthquake ground-response analysis, liquefaction assessment, and soil dynamics problems [8,9,10,11].

The shear wave velocity and small-strain shear modulus of soil can be directly determined by using sensors or testing apparatuses. Conventional laboratory testing methods such as resonant column tests and quasi-static loading tests (e.g., cyclic torsional shear and triaxial tests) have been applied to assess the shear wave velocity or the small-strain shear modulus. Resonant column tests are known as the most reliable technique for measuring the small-strain shear modulus [12]. In resonant column tests, the shear wave velocity is calculated from the fundamental frequency, mass polar moment of inertia of the soil specimen, inertial toque, and height of the soil specimen [2]. Cyclic torsional shear tests are generally conducted using the resonant column apparatus. The small-strain shear modulus is calculated from the steepest slope at the small strain of the backbone curve, which forms a basis for the stress–strain response. However, the driving system for exciting torsional load and the motion monitoring system of the resonant column apparatus are attached to the soil specimen. This setup changes the boundary condition of the soil specimen, thereby resulting in errors in the interpretation of test results [13]. Also, the tests that use the resonant column apparatus are generally expensive, highly sophisticated, and time consuming [5]. In cyclic triaxial tests, the small-strain shear modulus is calculated from the stress–strain response in the same way as the cyclic torsional shear tests. However, the small-strain shear modulus obtained from cyclic triaxial tests is rarely used due to the low accuracy of cyclic triaxial tests in small-strain ranges [14]. The soil conditions are adjustable in the laboratory tests, which is why the parameters that affect the shear wave velocity and the small-strain shear modulus can be studied for producing more comprehensive results compared with in-situ tests [15]. However, the small-strain shear modulus obtained from the laboratory tests is generally lower than that obtained from the field due to the effects of soil disturbance; the sampling process causes a reduction in soil stiffness by weakening soil structure and cementation [16,17,18,19].

Field tests are conducted under in-situ ground conditions, and sampling, which results in soil disturbance, is not required [11]. Thus, measurements of in-situ shear wave velocity and small-strain shear modulus are preferred, although they are generally more costly and time consuming compared with laboratory tests [11,20]. Field tests for assessing the shear wave velocity (or small-strain shear modulus) can be divided into invasive and noninvasive methods. Among invasive methods, cross-hole seismic tests are widely used because the attenuation of the shear wave according to depth is not issued and the shear wave velocity determined from the cross-hole seismic tests is often considered a reference standard, unlike in-situ shear wave velocities determined from other methods [11,21,22]. However, cross-hole seismic tests are known to be slow, time consuming, and very expensive due to the need for multiple boreholes [11]. The spectral analysis of surface waves (SASW) method is one of the most commonly used non-invasive methods to evaluate the shear wave velocity without boreholesand is non-destructive, reliable, and inexpensive [23]. In the SASW method, dispersive characteristics of Rayleigh waves are utilized to investigate the variation in the shear wave velocity with depth [24,25,26,27]; the shear wave velocity profile is obtained by using an inversion algorithm and by comparing field and theoretical dispersion curves. The SASW method is useful at sites where drilling and soil sampling are restricted. However, the SASW method has some reported disadvantages [22,26,28,29]: an inversion process to obtain the shear wave velocity profile is greatly dependent on the experience; it is time consuming and labor intensive; professional skills are required for the interpretation; and evaluation of the shear wave velocity is difficult in irregularly layered soil, where the soil stiffness does not increase with depth.

A bender element (BE) is widely used as an alternative technique to overcome the limitations of conventional laboratory and field tests for measuring the shear wave velocity or the small-strain shear modulus after Shirley and Hampton [30]. BE is a piezoelectrical transducer composed of two layers of piezoceramic plates. A cross-sectional polarization is induced by applying voltage results in the bending of BE, and then shear waves are generated, and vice versa. With the use of a pair of BEs, the shear wave velocity can be determined as the ratio of the tip-to-tip distance between BEs and the travel time of shear waves transmitted through the soil specimen. BEs have been incorporated into various geotechnical instrumentations. Youn et al. [9] incorporated BEs into resonant column and torsional shear testing equipment, and they compared the values of the shear wave velocities obtained from the resonant column test, torsional shear test, and BEs. Lee et al. [31,32] suggested a new oedometer cell with BEs to evaluate both the compressibility of soils and the shear wave velocities under the zero-lateral strain condition. Byun et al. [33,34] showed that the BE could be used to monitor the stiffness characteristic of soils during direct shear tests. Furthermore, BEs have been utilized to characterize the stiffness of various mixtures, such as salt-cemented granular soils and controlled low-strength material [35,36]. Lee et al. [37] developed a penetration-type field velocity probe with BEs to evaluate the in-situ shear wave velocity according to depth. BEs have been used in both laboratory and field to evaluate the localized stiffness characteristics of aggregate materials, including large-sized particles [38,39,40,41]. Although BEs are typically small, inexpensive, nondestructive, and feasible to set up in most geotechnical instrumentations [9,42,43], they should be embedded into the soil; doing so may disturb the sample [44].

The objective of this study is to develop a new shear wave monitoring system using a pair of piezoelectric ring bender (RB) and to evaluate its suitability in compacted soils compared with that of BE and ultrasonic transducer (UT). RB is introduced, and the specimen preparation of compacted soils is then described. The shear wave monitoring systems using three different transducers are explained. The shear wave velocity and resonant frequency determined from using the three different transducers are compared. The variation in shear wave velocity and shear modulus along with different dry unit weights and water contents is discussed.

## 2. Materials and Methods

### 2.1. Ring Bender

In this study, the RB, which can generate shear waves without penetrating the soil, is used as an alternative to conventional sensors for determining the shear wave velocity of soil. RB is a multilayered piezoelectric actuator, which has the shape of a thin ring with an outer diameter of 20 mm, an inner diameter of 4 mm, and a thickness of 1.25 mm, as shown in Figure 1a. The RB is composed of several thin layers of piezoelectric material, alternating with internal electrodes, and three external electrodes are located on the side of the RB. The internal electrodes are stacked and connected to positive, middle, and negative external electrodes. Voltages between +100 V and −100 V can be applied to the external electrodes [45]. The bending motion induced by the deformation of the RB can lead to the perturbation of the shear wave through the specimen without its disturbance. Note that the BE, which is commonly used as a shear wave transducer, needs to be embedded into a specimen.

The external electrodes of RB were connected to a coaxial cable (RG-316, VIMA Co., Ltd., Hwarihyeon, Hyangnam-Myun Whasung-City, Kyonggi Province, Korea) and Bayonet Neill–Concelman (BNC) connectors. Polyvinyl chloride (PVC) cement was used as a waterproof coating on the RB. According to the material properties reported by Montoya et al. [46], PVC cement is resistant to abrasion and has high coating integrity after being cured in 2 h at room temperature. The mechanical bond can be achieved by both roughing up the surface of the transducer and using a primer. Unlike epoxy, PVC cement can be uniformly distributed around the transducer and is flexible during vibrations. The hole inside the RB was filled with silicon to prevent soil particles from entering the sensor. The RB was fixed in an MC Nylon circular module with an external diameter of 24 mm, an internal diameter of 19 mm, and a thickness of 2.5 mm, as shown in Figure 1b). The RB was mounted at the edge of the circular module with epoxy glue. To avoid additional loss of performance, the bottom line of piezoelectric material in the RB should not be glued. The RB can be deformed both upward and downward.

### 2.2. Specimen Preparation

In this study, weathered soil sampled in Daegu, South Korea, was used for compacted specimens in monitoring shear waves according to the dry unit weight. On the basis of the sieve analysis, the grain size distribution of the soil is plotted in Figure 2a, and the index properties of the soil are summarized in Table 1. The mean diameter of the weathered soil was 0.87 mm, and the percent of the particles that passed through a No. 200 sieve was 2%. The gradation coefficient and the uniformity coefficient of weathered soil were 1.4 and 5.5, respectively. The liquid and plastic limits of the soil were 30.6 and 26.1%, respectively, and the plastic index was 4.5%. Based on the Unified Soil Classification System, the weathered soil was classified as poorly graded sand. According to the compaction curve reported by Byun and Kim [47], the optimum water content and maximum dry unit weight of the weathered soil were 9.8% and 20.6 kN/m^3^, respectively, as shown in Figure 2b. Considering the value of the liquid limit of the used soil and the results reported in a previous study [48], a typical single-peak compaction curve was expected to appear.

In this study, five different water contents were selected for preparing the compacted specimens. Each specimen with water content was compacted by applying the procedure of the modified proctor test specified in ASTM D1557 [49]; this process is shown in Figure 3. First, each specimen was divided into five layers in a mold with a diameter of 150 mm and a height of 135 mm; the grain size distribution of the soil for each layer was the same as the result obtained from the sieve analysis. Then, 56 blows were applied to each layer by using a 43 N hammer falling from a height of 457 mm, as shown in Figure 3a. After the fifth layer was finished compacting (Figure 3a), the collar and the soil in the collar on the mold were removed, and then the surface of the specimen in the mold was flattened, as shown in Figure 3c. Finally, the bottom plate was disassembled from the mold to characterize waves transmitted through the specimen (Figure 3d). The dry unit weights of the five specimens prepared in this study ranged from 19.7 to 20.7 kN/m^3^, which were slightly higher than the compaction curve previously reported by Byun and Kim [47]. After the shear waves were measured, the water content of the compacted specimen was measured according to ASTM D2216 [50] once again, and the measured water contents and dry unit weights of the compacted specimens are summarized in Table 2.

### 2.3. Shear Wave Monitoring

Shear waves in the five compacted specimens were monitored by three different piezoelectric transducers. Figure 4 shows the structure of these piezoelectric transducers and the configurations with electronic devices for the measurement of shear waves in the compacted specimens. The RB introduced in this study was used to measure shear waves in the specimens. In addition, a pair of UTs (SWC75, Ultran) with a diameter of 25 mm and a height of 16 mm was used to measure shear waves. The diameter of piezoelectric material in the UTs corresponds to 19 mm. To improve the coupling between the UT and the tested material, vacuum grease was applied to the surface of the matching layer [51]. As another shear wave transducer, the BEs, which are composed of a metal shim covered with piezoelectric materials on each side, were used. The BEs had a width and length of 20 mm, and the cantilever length of the BE was 3 mm.

The three transducers were located at the center of both surfaces of the compacted specimen. Shear wave measurements for the compacted specimens were performed according to the following procedure. First, the RB was applied to the surface of the compacted specimen to measure the shear waves. A pair of RBs was used as sender and receiver for shear waves on each surface of the specimen, respectively. Second, vacuum grease was applied to the surface of the UT. A pair of UTs was affixed to each surface of the specimens. Lastly, the shear waves were measured by using a pair of BEs. One of the tips of each BE was mounted on a module to transmit the shear wave. A 3 mm groove was made on both ends of the compacted specimen, and then the other tip of each BE was embedded into the surface of a specimen [52]. Considering the sample disturbance induced by BE, the RB and UT may be more useful for measuring the shear waves on the surface of the specimen. The shear wave transducer can also generate compressional waves, which can be reflected at the sidewall and may hinder the determination of the first arrival of shear waves [8,36]. Assuming that the Poisson’s ratios of the compacted specimens are greater than 0.13, the half-width of the mold used in this study is greater than the value of the critical boundary width proposed by Byun et al. [36]. To prevent any reflected waves from the ground, a pair of styrofoam supports was placed under the mold during the wave measurement.

The shear wave measurement system consists of a preamplifier, a signal generator, a filter-amplifier, and an oscilloscope.A step pulse with a voltage of 10 V and a frequency of 20 Hz is applied by using the signal generator, and the input signal can be magnified up to 200 V through the pre-amplifier. A filter amplifier is used to remove out the undesired frequency and to allow the signals of 500 Hz to 1 MHz to be received and magnified again. Finally, the output signal is saved through the oscilloscope after 1024 signals have been stacked.

## 3. Results and Discussion

### 3.1. Time-Domain Response

The signals obtained from the RBs at three different water contents are plotted in Figure 5a. The waveform of output signals varied according to water content in compacted soil. Figure 5b shows the variation in the first arrival time of the shear wave and the maximum amplitude of the output signal along with the water content. The time to zero voltage after the first bump was determined as the first arrival time of the shear wave [8]. The maximum amplitudes of the signals at water contents of 9.3 and 9.6% were greater than those at higher water contents. At water content ranging from 9.6 to 10.8%, the maximum amplitude of the signals decreased as the water content increased. The water content of 9.6% is the closest to the optimum water content of the compacted soils. The shortest first arrival time was found at the water content of 9.6%, and at the water content higher than 9.6%, the first arrival time increased with an increase in the water contents. Similar to the RB, the trends of the maximum amplitudes and first arrival times of signals obtained from the UT and the BE changed according to the water content of the compacted specimens.

The effect of the magnitude of the input signal on the waveform of the output signal was investigated. The voltage of input signals varied from 10 to 50 V through the pre-amplifier. The variation in output signals obtained from the RB according to the input voltage is plotted in Figure 6. As the magnitude of input signals increased, the maximum amplitude of the output signals gradually increased, while the first arrival time of the output signals remained constant regardless of the magnitude of the input signal. Accordingly, when the amplitude of the output signal obtained from the RB is small, the amplification of the input signal might be a useful method to obtain a clearer output signal and to accurately determine the first arrival time.

The response of the RB is maximized when the frequency of the sinusoidal input signal corresponds to the resonant frequency. Figure 7 shows the variation in the output signals obtained from the RB along with the frequency of the input signal at the water content of 9.6%. For the input frequencies of 1 and 2 kHz, the amplitudes of signals were more significant compared with other input frequencies. As the frequency of input signal increased, the amplitudes of low-frequency components of the output signals gradually decreased, whereas those of high-frequency components were more dominant compared with those of the low-frequency components. Accordingly, the first arrival times of shear waves varied along with the selected input frequency. Considering that the resonant frequency for the specimen with a water content of 9.6% was 1.35 kHz, the input frequencies above 2 kHz were not suitable for determining the first arrival time of the shear wave.

### 3.2. Frequency Domain Response

The frequency characteristics of the output signals obtained from the RB were analyzed by using Fourier transform. Figure 8 shows the evolution of frequency spectrums for the RB in the compacted specimens according to the water content. For the frequency spectrum of the RB, the specimen with the water content of 9.3% showed the highest resonant frequency, which is defined as the frequency where the fast Fourier transform (FFT) amplitude in the frequency domain is a relative maximum. The resonant frequency was reduced up to 0.85 kHz at the water content of 12.4%. The variation of resonant frequencies for the three transducers along the water content is plotted in Figure 9. The resonant frequencies of RB and BE decreased with an increase in the water content. The resonant frequency of UT fluctuated as the water content changed. At the water content of 9.3%, the resonant frequencies of RB and UT were 2.70 and 0.64 kHz, respectively; the resonant frequency of UT was clearly smaller than those of RB and BE. At other water contents, the difference in resonant frequencies among the three transducers was less than 0.55 kHz. With the assumption that the transducer installation condition remained constant, the resonant frequency of transducers depends on both the density and stiffness of the soil [8,53]. Thus, the increase in the resonant frequency of the transducers represents an increase in either soil density, stiffness, or both.

The resonant frequencies obtained from the RB were compared with those obtained from the UT and BE, as shown in Figure 10. The slope and the coefficient of determination of the linear relationship between the resonant frequencies of the RB and the BE were 0.995 and 0.845, respectively, when the y-intercept of the linear relationship corresponded to zero. For the linear relationship between the resonant frequencies of the RB and the UT, the linear trend line showed a negative slope and a value close to zero of the coefficient of determination. At the water content of 9.3%, the difference in resonant frequencies between the RB and the UT was significant. Thus, the data point in Figure 10 was far from the line of equality. In short, any meaningful relationship between the resonant frequencies of the RB and the UT could not be established due to the scattered data.

### 3.3. Shear Wave Velocity

The shear wave velocity can be determined from the first arrival time recorded on the output signal and the measured distance between a pair of transducers. The variation in shear wave velocities obtained from the three transducers along the water content of the specimens is plotted in Figure 11. Overall, the shear wave velocity decreased as the water content of the specimen increased. For UT and BE, the values of shear wave velocities at the water content of 9.3% were higher than those at water contents ranging from 9.6 to 12.4%. As for the RB, the shear wave velocity at the water content of 9.6% was greater than those at the other water contents. Compared with those of the BE and UT, the shear wave velocities of the RB were faster, except at the water content of 12.4%. The shear wave velocities of the RB were similar to those of the specimens reported by Indraratna et al. [54] and Heitor et al. [55], as indicated by the gray area in Figure 11. According to previous studies [54,55], the values of shear wave velocity obtained from the compacted silty sands increased with the compaction energy, and those obtained at wet of optimum decreased with an increase in the water content. For dry of optimum in particular, the shear wave velocity at the lowest water content can be greater than that at a higher water content. These results mean that the shear wave velocity at dry of optimum may be significantly influenced by the soil density as well as the water content. Figure 12 shows a comparison of the shear wave velocity of the RB with those of the UT and BE. Overall, the shear wave velocities of the RB were faster than those of the UT and BE, showing that the slopes for the linear relationships of shear wave velocities for both RB‒UT and RB‒BE were approximately 1.3. The coefficient of determination for the shear wave velocity relationships of RB‒UT and RB‒BE ranged from 0.725 to 0.813.

The relationship between the dry unit weight of specimens and the shear wave velocity at each water content is plotted in Figure 13. For the three transducers, the shear wave velocity generally increased with the dry unit weight of the specimen. The shear wave velocity (*V_s_*) can be represented by the dry unit weight (*γ_d_*) as follows:(1)$$V_{s}\;=\;\alpha \cdot \exp\left(\beta \gamma_{d}\right)$$
where *α* and *β* are constants determined by regression analysis. The values of *α* and *β* for each transducer are summarized in Table 3. For the RB, the coefficient of determination was greater than those for the UT and BE. Considering that the maximum dry unit weight and maximum shear wave velocity estimated by the RB were obtained at the water content of 9.6%, the RB may be more influenced by the effect of soil density compared with the BE and UT. The values of dry unit weights estimated from using the three transducers were lower than those estimated from using empirical relationships suggested by previous studies [56,57,58]; these empirical relationships were established based on field testing data.

### 3.4. Small-Strain Shear Modulus

The small-strain shear modulus determined from the soil density multiplied by the square of shear wave velocity is plotted in Figure 14. The relationships between the shear modulus and water content were similar to those between the shear wave velocity and water content. Overall, the small-strain shear modulus decreased with an increase in the water content, except for the RB at dry of optimum. At the water content of 9.6%, the value of the small-strain shear modulus estimated from the RB was maximized. The maximum values of small-strain shear modulus estimated from the UT and BE were 150 and 162 MPa, respectively. The minimum values of small-strain shear modulus estimated from the three transducers ranged from 22.5 to 41.1 MPa. Heitor et al. [59] showed that the compacted specimens had a small-strain shear modulus of about 100 MPa at dry of optimum and a small-strain shear modulus of 20 to 40 MPa at wet of optimum. The soil used by Heitor et al. [59] was classified as SP-SC.

## 4. Conclusions

A new shear wave monitoring system that uses a pair of RBs was proposed in this study. The RB is a multilayered piezoelectric transducer that generates shear waves through bending motion. The RB was connected to the coaxial cable and BNC, waterproofed with PVC cement, and mounted on a module. To assess the suitability of the RB, five compacted specimens were prepared with different water contents, and the shear waves were measured using three different shear wave transducers: RB, UT, and BE. The measurement system for the shear wave transducers consisted of a preamplifier, a signal generator, a filter-amplifier, and an oscilloscope.

The time-domain responses for the five compacted specimens were analyzed to evaluate the effects of input voltage and frequency of the transducers on the shear wave velocities. The experimental results demonstrated that the waveforms of the output signals measured from the RB varied according to the water content of the specimen, while the first arrival time of the output signals remained constant along the magnitude of the input signals. For the sinusoidal input signal of RB, the first arrival time of the output signal was influenced by the frequency of the input signal. An analysis of the frequency domain response found that the difference in the resonant frequencies between the three transducers was less significant, except for that at the water content of 9.3%. Overall, at wet of optimum, the shear wave velocities and small-strain shear moduli for the three transducers decreased as the water content increased. The shear wave velocities for the RB were slightly greater than those for the other transducers. The RB showed the exponential relationship between the shear wave velocity and dry unit weight with a higher coefficient of determination better than the UT and BE did. The RB may be one of the effective shear wave transducers for evaluating the shear wave velocity and small-strain shear modulus of compacted soils.

## Figures and Tables

**Figure 1 sensors-21-01226-f001:**
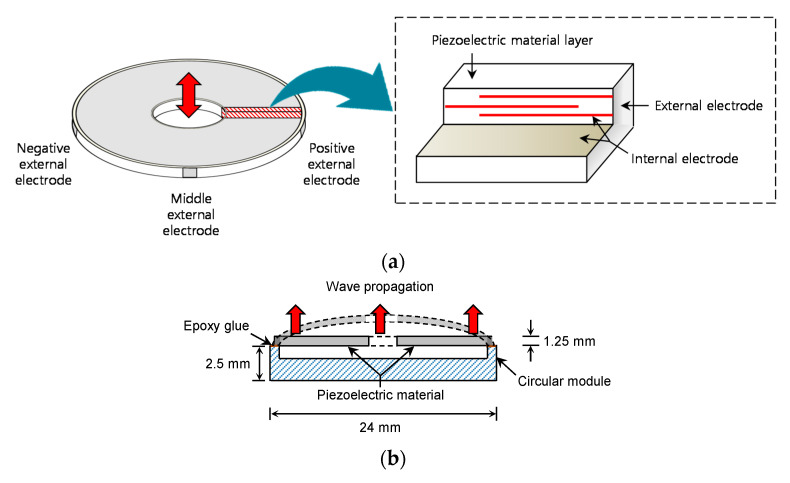
Schematic of ring bender: (**a**) shape and components; (**b**) installation.

**Figure 2 sensors-21-01226-f002:**
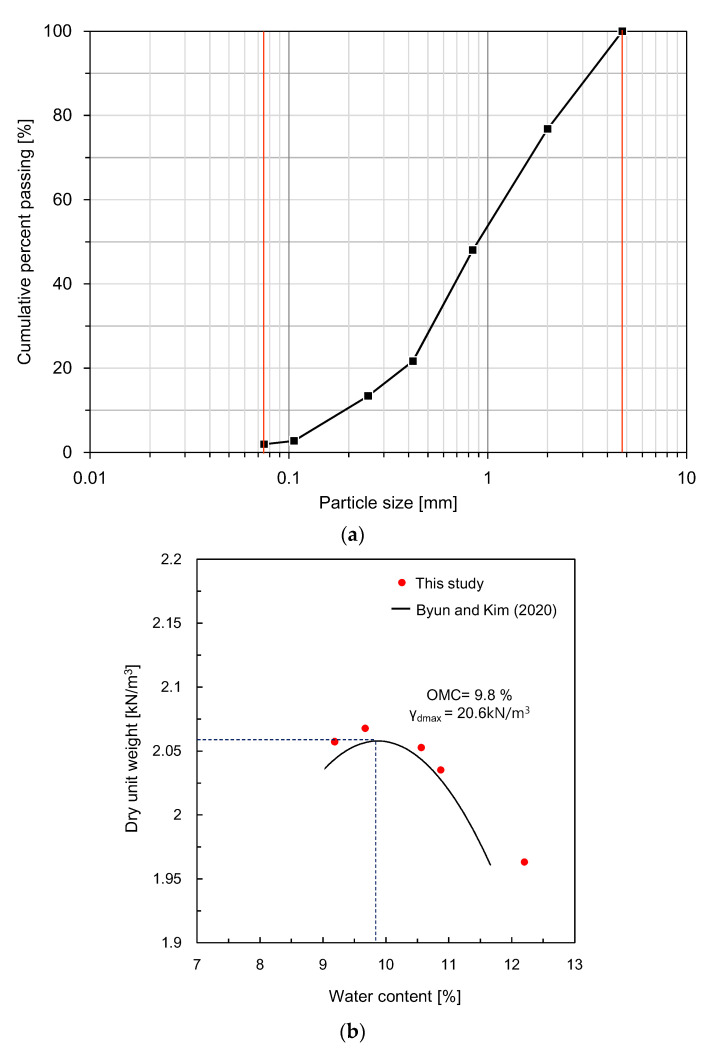
Properties of the compacted soils: (**a**) particle size distribution; (**b**) compaction curve.

**Figure 3 sensors-21-01226-f003:**
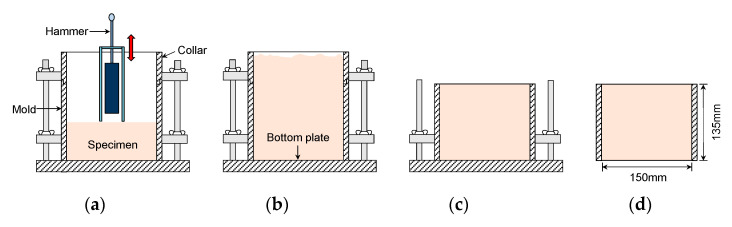
Procedure for compacting specimens: (**a**) compaction; (**b**) filling completion; (**c**) removal of collar; (**d**) removal of bottom plate.

**Figure 4 sensors-21-01226-f004:**
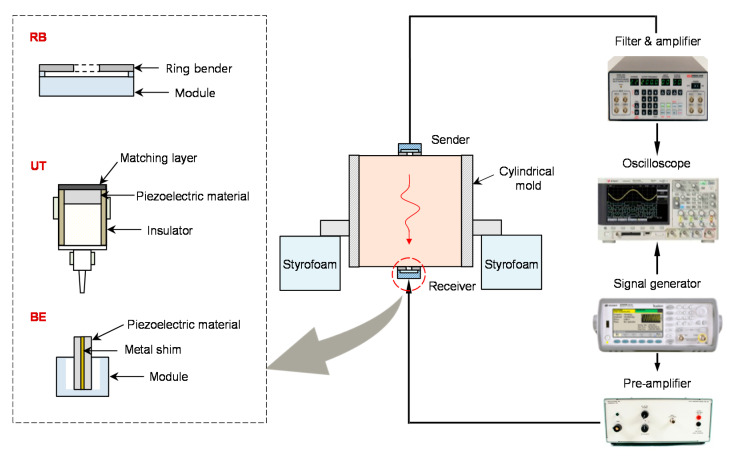
Schematic of the experimental setup for the measurement of shear waves.

**Figure 5 sensors-21-01226-f005:**
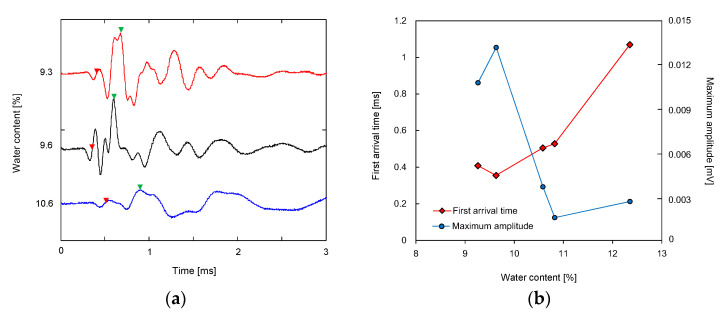
Results obtained from ring bender: (**a**) waveforms at different water contents; (**b**) variation in first arrival time and maximum amplitude with water contents. Red and green triangles denote the first arrival time and the time to maximum amplitude, respectively.

**Figure 6 sensors-21-01226-f006:**
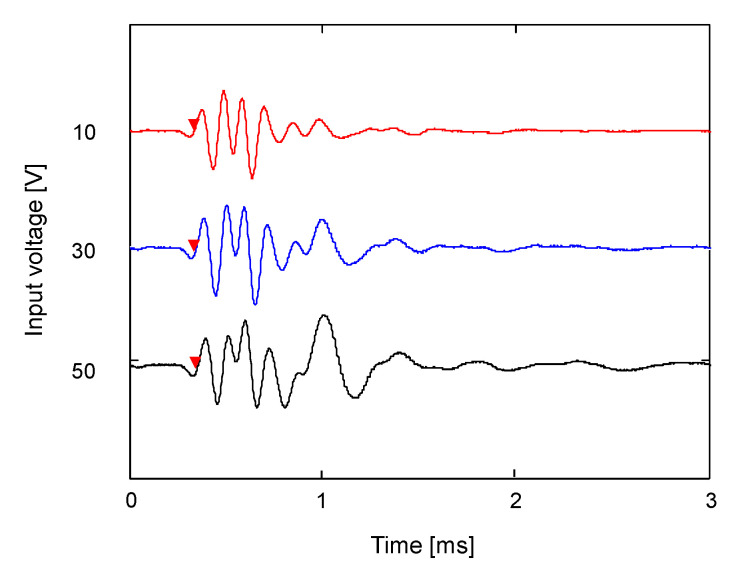
Variation in waveform along with the input voltage for ring bender. The red triangle denotes the first arrival time.

**Figure 7 sensors-21-01226-f007:**
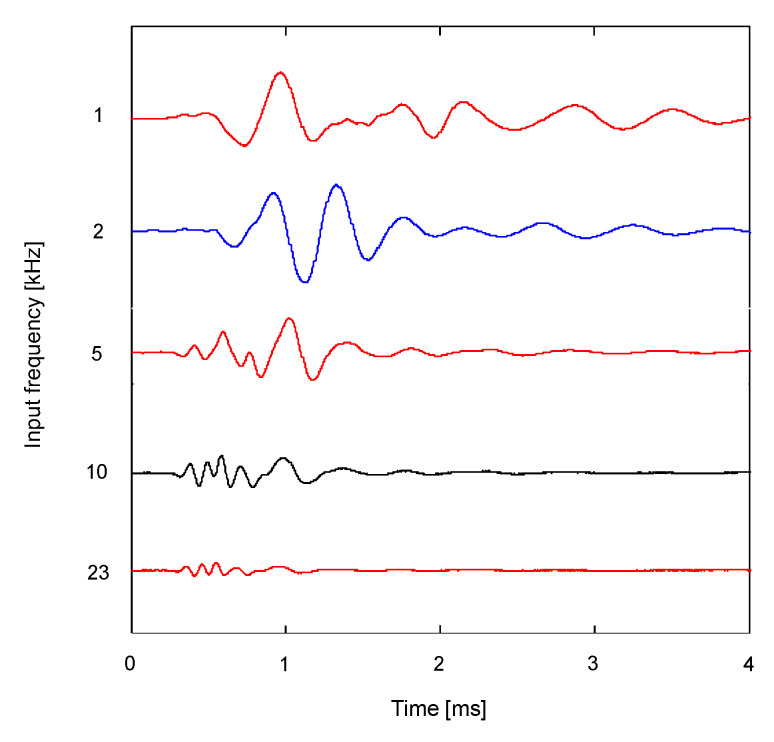
Variation in waveform obtained from ring bender along with the frequency of input signal.

**Figure 8 sensors-21-01226-f008:**
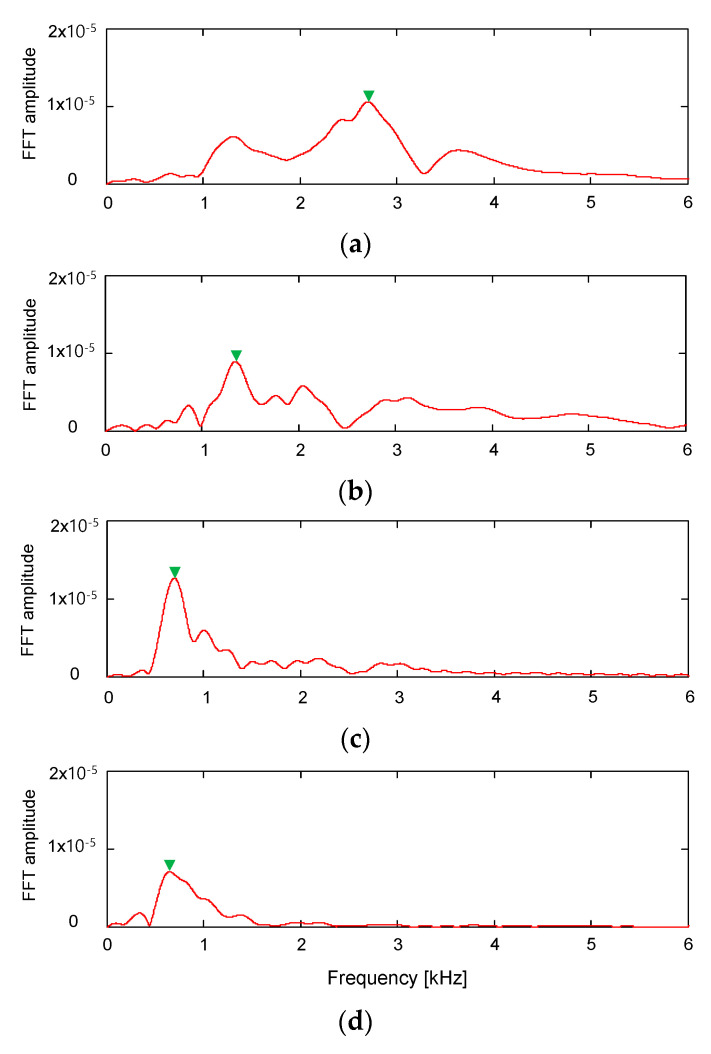
Frequency response curves obtained from the ring bender at the water content of (**a**) 9.3%; (**b**) 9.6%; (**c**) 10.6%; (**d**) 12.4%. FFT denotes fast Fourier transform.

**Figure 9 sensors-21-01226-f009:**
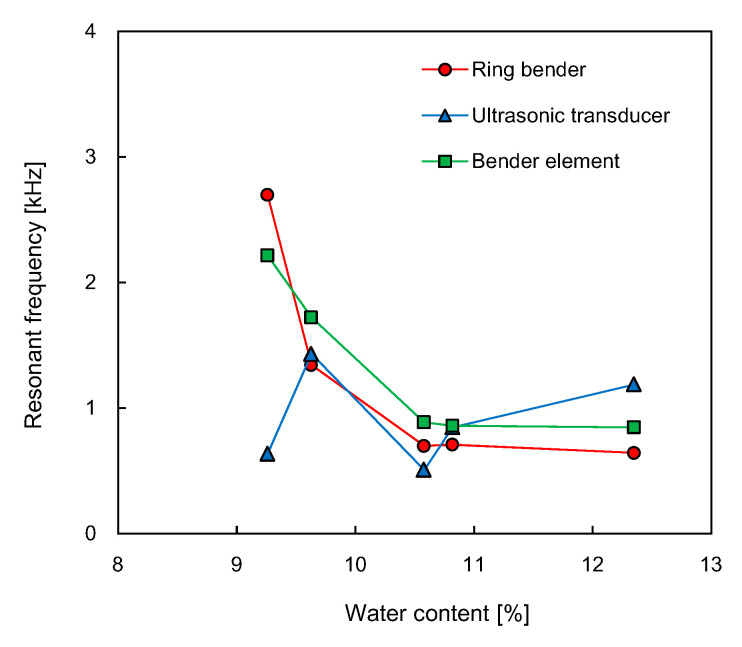
Variation in the resonant frequency of shear waves estimated from the three different sensors.

**Figure 10 sensors-21-01226-f010:**
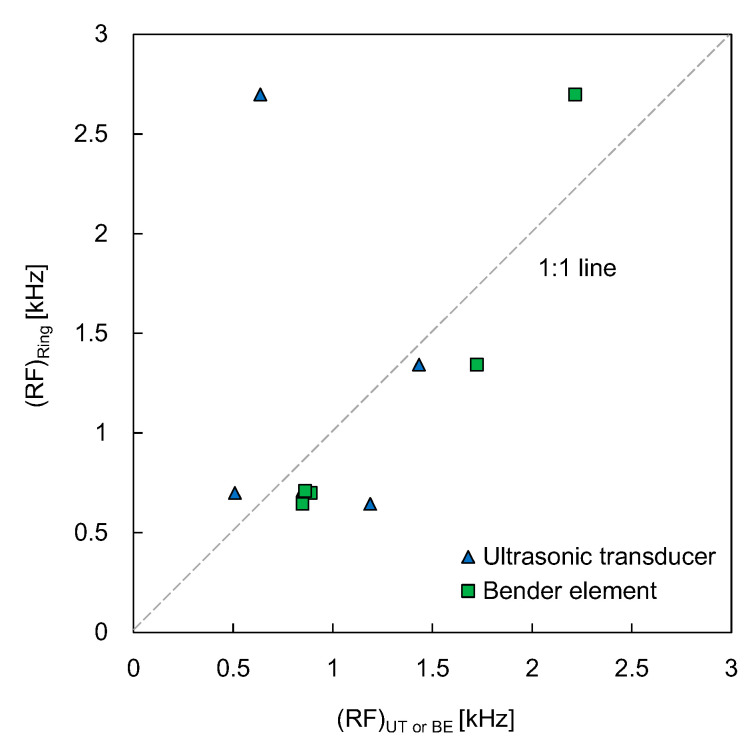
Comparison of resonant frequency obtained from the ring bender with ultrasonic transducer and bender element. RF denotes the resonant frequency.

**Figure 11 sensors-21-01226-f011:**
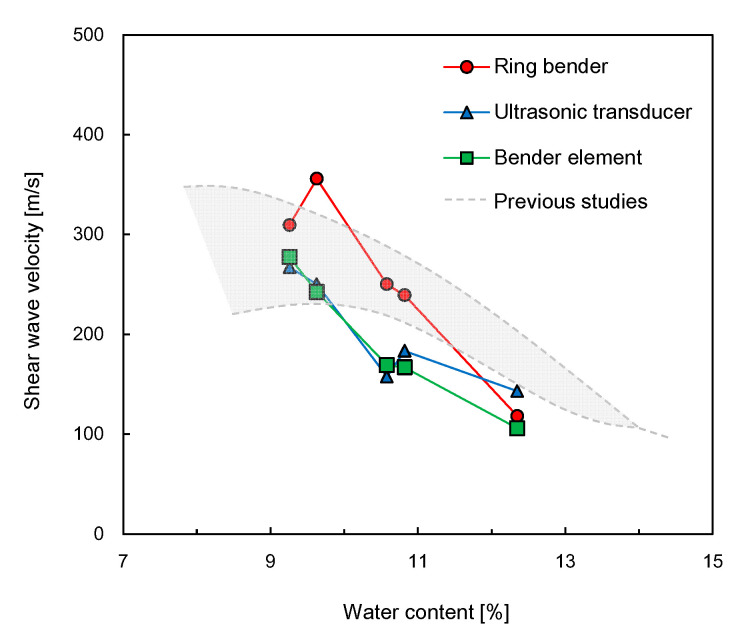
Relationship between water content and shear wave velocity.

**Figure 12 sensors-21-01226-f012:**
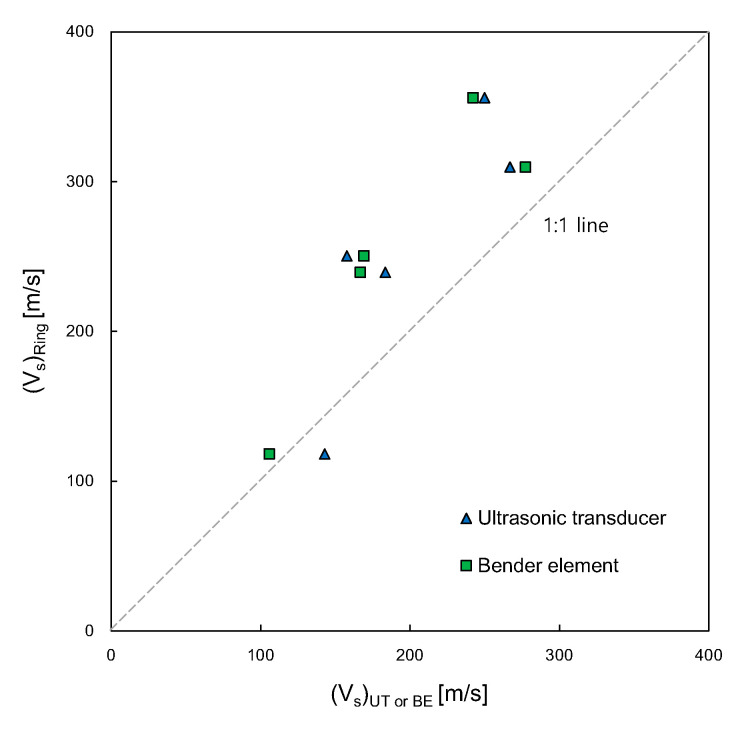
Comparison of shear wave velocities obtained from the ring bender with ultrasonic transducer and bender element.

**Figure 13 sensors-21-01226-f013:**
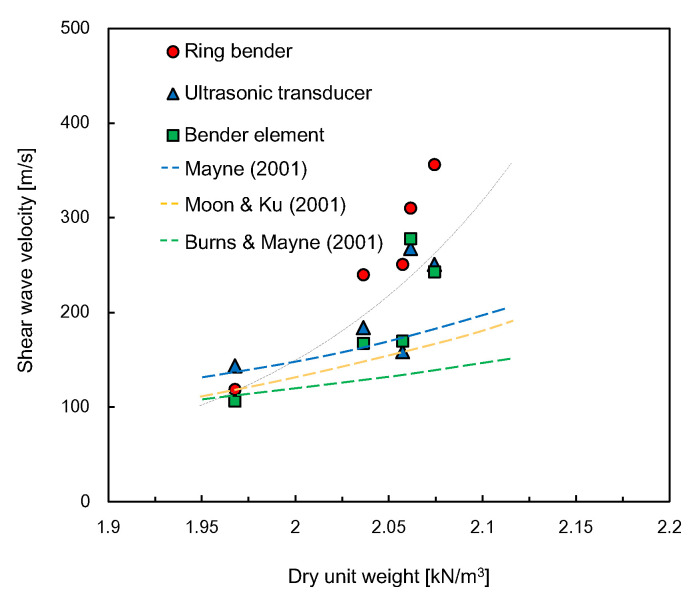
Relationship between dry unit weight and shear wave velocity.

**Figure 14 sensors-21-01226-f014:**
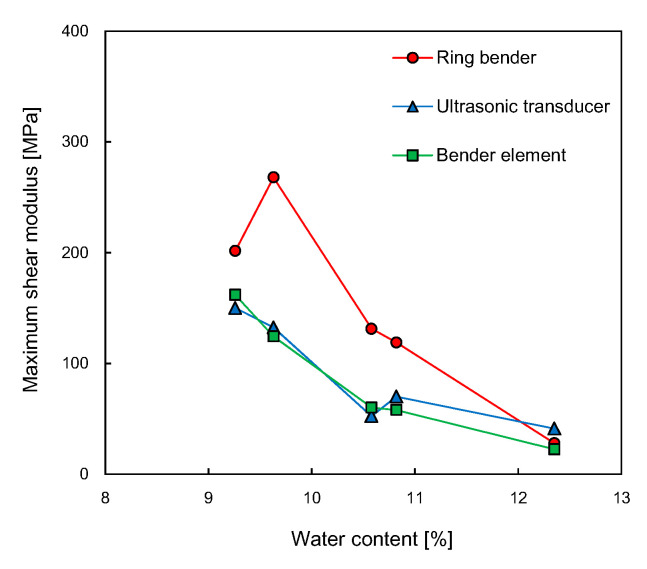
Relationship between water content and maximum shear modulus.

**Table 1 sensors-21-01226-t001:** Index properties of the soil used in this study.

Specific Gravity G_s_	Grain Diameters to a Percent Passing [mm]				Gradation Coefficient C_c_	Uniformity Coefficient C_u_	Unified Soil Classification System
	D_10_	D_30_	D_50_	D_60_			
2.66	0.19	0.52	0.87	1.05	1.4	5.5	SP

**Table 2 sensors-21-01226-t002:** Water content according to dry unit weight.

**Dry Unit Weight [kN/m^3^]**	20.6	20.7	20.6	20.4	19.7
**Water Content [%]**	9.3	9.6	10.6	10.8	12.4

**Table 3 sensors-21-01226-t003:** Constants for the relationship between shear wave velocity and dry unit weight.

	α	β	R^2^
**Ring Bender**	4 × 10^−7^	9.886	0.969
**Ultrasonic Transducer**	0.0115	4.773	0.537
**Bender Element**	2 × 10^−5^	7.956	0.804
**Entire System**	4 × 10^−5^	7.539	0.682

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
