# Peer review of "Piezoelectric Ring Bender for Characterization of Shear Waves in Compacted Sandy Soils"

_sensors, 2021, doi:10.3390/s21041226_

Round 1

Reviewer 1 Report

A new shear wave monitoring system using a pair of RBs is proposed by the authors. The RB consists of a multilayered piezoelectric transducer that generates shear waves through bending motion. Different tests and parametric studies are conducted.

The paper is of interest, well written, and deserves attention. Prior to any application, the authors should do the following:

Check the manuscript once more w.r.t English language, e.g. by Grammar checking tools like Grammarly or by native speakers. There are not many issues, but still ...

The work presented here is merely an experimental work. The authors may comment on which extend by means of numerical simulation, the design of the testing system could be systematically optimized. For this, the following works might be of value

Numerical Simulation of Mechatronic Sensors and Actuators, Manfred Kaltenbacher

Forward and Inverse Problems in Piezoelectricity, T. Lahmer

Inverse piezoelectric material parameter characterization using a single disc-shaped specimen, Feldmann et al

or related

All graphs should be embedded as PDFs in order to guarantee full scalability and thus high quality of the figures

Reviewer 2 Report

Dear Authors,

I would like to appreciate the respected authors for their research work to assess the effect of applicated various methods to obtain the shear wave velocity in granular soil, using BE, RB and UT tests. There are a lot of comments on the prepared manuscript that must be explained or clarified in more detail in the text and figures of the paper. I recommend to the authors to reread it once more to improve it before the recomendation to publishing or not. Some of my main comments are listed as following:

Some comments:

First very important question. Is it paper registered as manuscript or technical note?

Abstract.

In Abstract You mentioned about time- domain response, but from presented in Chapter 3 Results and Disscussion, 3.1 Time Domain Response it is not is clear. Very weak presentation on figures, no explanation on figures, nor correlation between text and figures.  Could You explain more the obtained results, and how interpretation of Time Domain response was done. It is not clear for me.

1 - Introduction. Maerials and Methods.

In the Introduction You wrote that „BEs should be embedded into soil, which may disturb the samples”.

In the Materials and Methods only instalation of Ring Bender is widely explained.

The RB, UT, BE measures was done on the same samples. What was the turn of measurements. What was first, the second and the third. Propably You started with UT measurements, than with RB tests, at the end was BE method, but how the Bender Elements was instaled in grain material with maximum dimension 4 mm (Figure 2 a). Am I right? Please claryfy this.

Figure 1 is not clear for me. You have a hole inside the RB, but Wave propagation is coming from whole surfaces. Am I right? Please explain this. Also did You discuss the possible problem of measurement disturbances in a steel cylinder?

Figure 2 a. Please add the vertical barriers for sand fractions, minimum and maximum value. I recognize that material contain about 20% grain more than 2 mm up to 4 mm. According to Eurocode 7 (european codes) that material should be describe as sandy gravel or gravel, of course poorly graded. From grain size distribution curve, we now not to much about fraction less than 0.063 mm. Less than0.063 mm it is silt fraction, How many percent – about 3-4 percent. It is important for me to discuss Figure 2b – modified Proctor Curve.

Figure 2b. Do You know the manuscript Lee P.Y., Suedkamp R.J. ( 1972), titled „Characteristics of irregular shaped compaction curves of soils”. Highway Research Recorg 381, National Academy of Science. Washington, 1-9.? From the manuscript You can get to know about different shape – type of compaction curve. Presented Compacted Furve fits to Type A, but it is representative for cohesive material. For sandy material should be Type B (with named 1.5 peak) or Type C (2 peaks) It depends of minerals  found in sands. Presented by You values of Dry Unit Weight are higher thah shoul be for sands, and are typical for gravel grains.  Proctor Curve also shows the water sensitivity as for cohesive soils, as for sandy clay. The water content – 12.4% is not possible to preapare a noncohesive soil sample ready to test. It is wery wet and propably was not possible to properly compact.

Figure 2b. Please ad to figure optimum water content and maximum dry unit density.

I read in text that is weathered material, but could you discuss with my interpretations and give an appropriate explanation.

Figure 3. No explanation for Figure 3d. Please complete.

Figure 4. Schemating driwing is clear. But I would like to see real pictures from tests with instaled transducers: RB, UT, BE. Is it possible.

Why the smples were compacted verically but test was done horizontaly. Is it clear for interpretation the results of the tests”?

Figures 5, 6,7 are not clear. As I mentioned before mothing is coming from figures. The interpretation in text not correlates with presented results. Please improve quality of figures.

 Figures 8 a, b c, d, e. Please provide a description of the vertical axis.

Interpretation of results contains more conclusions than real interpretation. Please reread and improve.

Figures 9 and 10 are not satisfactory because of my interpretative reservations previously given.

Figure 10 need descripion of presented points – see explanation on Figure 9.

I believe that the number of samples made is not enough for the content of the conclusions.

There is no statistical analysis in the article that would allow to prove the analyzes performed. They are too general.

Did the authors carry out comparative tests for their own use using a resonance column or tests in a triaxial compression apparatus equipped with bender elements?

I believe it would be very appropriate to prove the statements in the conclusion.

I am asking the authors to comment on my remarks. I consider it necessary for further discussion of the presented material before the final decision.

As mentioned before, the accuracy of the explanation depends on whether it is a manuscript or a technical note.

Yours faithfully.

Rewiever.

Round 2

Reviewer 2 Report

As I said previously I like that paper. The application of new technics is always desired. I am satisfied from authors response. Now I am waiting for next paper - results with comparative tests.

Best regards

Reviewer